# Rapid Prediction of Mechanical Properties Based on the Chemical Components of Windmill Palm Fiber

**DOI:** 10.3390/ma15144989

**Published:** 2022-07-18

**Authors:** Liyuan Guan, Qiuzi Huang, Xiaoju Wang, Ning Qi, Mingxing Wang, Guohe Wang, Zhong Wang

**Affiliations:** 1College of Textile and Clothing Engineering, Soochow University, Suzhou 215006, China; 20174215007@stu.suda.edu.cn (L.G.); 20205215012@stu.suda.edu.cn (Q.H.); 20184015015@stu.suda.edu.cn (X.W.); 2National Engineering Laboratory for Modern Silk, Soochow University, Suzhou 215123, China; qining@suda.edu.cn; 3Violet Home Textile Science and Technology Co., Ltd., Nantong 226311, China; wangmingxingking@163.com; 4China National Textile and Apparel Council Key Laboratory of Natural Dyes, Soochow University, Suzhou 215123, China

**Keywords:** windmill palm fiber, chemical component, near infrared spectroscopy, Young’s modulus, grey model

## Abstract

During spinning, the chemical component content of natural fibers has a great influence on the mechanical properties. How to rapidly and accurately measure these properties has become the focus of the industry. In this work, a grey model (GM) for rapid and accurate prediction of the mechanical properties of windmill palm fiber (WPF) was established to explore the effect of chemical component content on the Young’s modulus. The chemical component content of cellulose, hemicellulose, and lignin in WPF was studied using near-infrared (NIR) spectroscopy, and an NIR prediction model was established, with the measured chemical values as the control. The value of R_C_ and R_CV_ were more than 0.9, while the values of RMSEC and RMSEP were less than 1, which reflected the excellent accuracy of the NIR model. External validation and a two-tailed *t*-test were used to evaluate the accuracy of the NIR model prediction results. The GM(1,4) model of WPF chemical components and the Young’s modulus was established. The model indicated that the increase in cellulose and lignin content could promote the increase in the Young’s modulus, while the increase in hemicellulose content inhibited it. The establishment of the two models provides a theoretical basis for evaluating whether WPF can be used in spinning, which is convenient for the selection of spinning fibers in practical application.

## 1. Introduction

Windmill palm, found in tropical and subtropical regions, is a lignocellulosic tree species that can provide renewable biomass; it also exhibits strong resistance and absorption capacity to various harmful gases, such as smoke, sulfur dioxide, and hydrogen fluoride. In recent years, China’s promotion of millions of acres of windmill palm tree planting land has greatly increased the annual output of windmill palm fiber (WPF). Leaf sheaths, fruit bunches, petioles, trunks, palm sheath meshes, and WPF peeled from these have been widely used in the production of ropes, furniture, filter materials, and packaging materials due to their excellent hardness, porosity, and natural degradability [1]. With further studies on WPF, its application in the textile field has broad prospects, such as its use in textiles as an elastic composite material [1,2]. The mechanical properties of WPF have a great influence on the performance of textiles, which in turn is affected by the content of chemical components [3,4]. According to previous studies, there are certain differences in the chemical component content of fibers extracted from various parts of the WPF (Table 1) [5,6,7]. Therefore, in order to meet the need for the rapid selection of spinning fibers in the industrial production process, the chemical component content of WPF should be determined as soon as possible, and the relationship between the two should be discussed.

Traditional wet chemical analysis methods, such as the quantitative analysis of ramie chemical components, the Van Soest method, and volumetric analysis have the characteristics of being destructive, laborious, time-consuming, and environmentally unfriendly [8], which can no longer meet the development trend of modern analytical chemistry towards speed, convenience, and time-conservation. In this case, near-infrared (NIR) spectroscopy, which has many advantages, such as non-destruction, fast analysis speed (about 3 min), wide application range, and simple environmental protection [9], has been widely used in food [10], agriculture [11,12], industry [13,14], medical [15], and other fields. This technology is not only chemical reagents free, but can also quickly analyze multiple components at the same time, which greatly improves test efficiency, reduces the cost of enterprises, and reduces the pollution to the environment, which is in line with the trend of modernization and green development of the industry. Therefore, NIR spectroscopy may be a better choice for quantitative analysis of fiber chemical components [16,17]. Furthermore, the grey model (GM), proposed by Deng Julong, can effectively deal with the uncertain data, explore the potential laws between variables, and then conduct systematic analysis and accurate prediction [18]. Compared with other prediction models, such as the artificial neural network, GM does not exhibit problems such as over-fitting and local minima, and is based on grey system theory [19,20]. Compared with the mechanical test method mentioned in ASTM C1557, GM is comprised of a simple and fast test, without requiring a lot of manpower, equipment, or other resources. Therefore, GM could be utilized to quickly determine the mechanical properties of WPF, and the relationship between the component content and mechanical properties could be explored by combining NIR with GM. GM has also developed rapidly due to its high prediction accuracy, less lower data requirements for modeling, and its ability to use disordered data; it has been widely used in the machinery, transportation, environment, and materials industries. Tangkuman and Yang et. al. [21] developed a grey model for predicting machine degradation; Xiao et al. [22] proposed a grey model of traffic flow in a road section based on a traffic flow dynamics theory to solve the problem of real-time dynamic traffic flow prediction. Ye et al. [23] proposed a grey model to measure the accumulating CO_2_ emissions impact on China’s transportation sector, and the predictive results were conducive to making recommendations for reducing future CO_2_ emission. Wang et al. [24] proposed an improved multivariate grey model (IMGM) for battery health prognostics, which could improve the reliability and efficiency of energy storage technologies.

In this case, the quick determination of the chemical components of the raw materials such as cellulose, hemicellulose, and lignin, may be of key importance for the repeatability of production. In addition, the chemical component content has a large impact on the strength of the products made from plant materials. After all, lignocellulose is the main component of many by-products of agri-food processing. Unfortunately, there are few reports on the rapid determination of mechanical properties of WPF by using a grey model, especially the precise determination based on the up to date chemical contents of fibers. At present, the grey model is simply used to predict the fineness of ramie fiber in the textile field [25].

The main purpose of this work is to develop a grey model for the rapid determination of mechanical properties based on the chemical components of WPF. Firstly, the NIR model was established to predict the content of each component quickly, and then the GM(1,4) model was established to realize the efficient prediction of the Young’s modulus of WPF. The rapid selection of fibers for spinning has promoted the industrialized production of WPF, which is conducive to the accelerated development of the transformation and upgrading of these enterprises.

## 2. Materials and Methods

### 2.1. Materials

The windmill palm sheath meshes, petioles, leaf sheaths, and fruit bunches were purchased from the Yuanmu Company, Wuhan, China. NaOH (CP) and H_2_O_2_ (30%,) were purchased from Guoyao Company, Beijing, China. Sodium hypochlorite (CP), glacial acetic acid (AR), and methanol (AR) were purchased from Jiangsu qiangsheng functional chemistry Co., Ltd., Suzhou, China. Sulfuric acid (AR) was purchased from Soochow University experimental material supply center.

### 2.2. Experimental Design

Different degrees of degumming were performed on WPF, with reference to previous research results, and were numbered 1–20 [26]. To make the model more adaptable, three parts of the petioles, leaf sheaths, and fruit bundles, with large differences in the components of windmill palm trees, were also selected., and the above three samples and the fibers extracted from them were numbered 21–26, respectively. The preparation methods of petioles fiber, leaf sheaths fiber, and fruit bunch fiber can be found in a previous paper [27].

### 2.3. Chemical Component Content Test

The contents of cellulose, hemicellulose, and lignin were determined by wet chemical analysis, with reference to previous experiments [28]. Prior to the test of chemical components, all windmill palm samples were dried for 24 h at 60 °C to eliminate moisture.

### 2.4. NIR Spectroscopy

The near-infrared spectrum of each sample was measured by a UV-Vis near-infrared spectrophotometer (Carry 5000, Agilent Technologies, Santa Clara, CA, USA). Each sample was sheared into powder and dried in an oven at 60 °C for 12 h for testing. Absorbance values were collected over the wavelength range of 4000 cm^−1^–12,000 cm^−1^ at intervals of 2 nm, with a scanning velocity of 3000 cm^−1^/min and an acquisition time of 0.1 s. The spectrum of three replicate measurements for each sample were averaged using OPUS infrared processing software (Bruker Corporation, Karlsruhe, Germany).

### 2.5. Young’s Modulus Measurement

According to ASTM C1557, the tensile property of the single fibers of samples No. 1–15 were measured by employing a universal material tester (Instron 5967, Instron, Boston, MA, USA) using a 500 N load cell with a 0.05 mm/s cross-head speed. Each group of samples was tested under three gauge lengths of 20 mm, 30 mm, and 40 mm, and each length was tested 15 times; scilicet, each sample was tested 45 times. The samples were dehumidified for 16 h under the desired conditions of temperature (20 ± 2) °C and relative humidity (65 ± 5)% before the tensile test. Young’s modulus could be calculated according to Equation (1) [26].
(1)$$\frac{\Delta L}{F}=\frac{L_{0}}{AE}+C_{s}$$
where Δ*L* refers to the total measured displacement; *F* represents the breaking strength; *C*_S_ represents the system compliance; *L*_0_ is the gauge length; *A* is the cross-sectional area of the fiber, and *E* is Young’s modulus. The cross-sectional area of the tensile section after the tensile property test was measured by a digital microscope (VHX-100, Keyence, Osaka, Japan). Therefore, in a linear representation of Δ*L*/*F* as a function of *L*_0_/*A*, Young’s modulus is the inverse of the slope, with system compliance as the intercept [29,30].

### 2.6. Parameter Definition of GM Establishment

The cellulose, hemicellulose, and lignin contents of samples No. 1–15, obtained through chemical component content testing, were taken as comparative sequences 1, 2, and 3, respectively, and Young’s modulus as reference sequence *X*_0_^(*m*)^(*k*), and the grey model of the chemical components and the mechanical properties of WPF was established. The grey differential equation of GM (1,4) was shown in Equation (2) [31].
(2)$$X_{0}^{\left(0\right)}\left(k\right)=b_{1}X_{1}^{\left(1\right)}\left(k\right)+b_{2}X_{2}^{\left(1\right)}\left(k\right)+b_{3}X_{3}^{\left(1\right)}\left(k\right)-aZ_{1}\left(k\right)$$
where the parameter *b_i_* (*i* = 1, 2, 3) represents the grey action quantity, with a as the development coefficient; *X*_0_^(0)^(*k*) refers to the initialization processing data; *X_i_*^(1)^(*k*) (*i* = 1, 2, 3) is the one-time accumulated generating date; and *Z*_1_(*k*) is the mean value generating date.

In order to reduce the discreteness of the data, the component contents and mechanical properties of each sample were initialized according to Equation (3) [32].
(3)$$X_{i}^{\left(0\right)}\left(k\right)={X_{i}}^{\left(m\right)}\left(k\right)/{X_{i}}^{\left(m\right)}\left(1\right)$$
where *k* refers to the number of rows.

The initialization data was processed according to Equations (4) and (5) [31], and the one-time accumulated generating operation sequence and the mean value generation sequence were obtained, respectively.
(4)$$X_{i}^{\left(1\right)}\left(k\right)=\sum_{i=1}^{k}X_{i}^{0}\left(k\right)$$
(5)$$Z_{1}\left(k\right)=0.5X_{0}^{\left(1\right)}\left(k+1\right)+0.5X_{0}^{\left(1\right)}\left(k\right)\cdots (k\geq 2)$$

The matrix *B* and the vector *Y_N_* were obtained through the mean value generation sequence and the one-time accumulated generating operation sequence, which could be expressed as:$$B=\left|\begin{array}{ccccc} Z_{1}\left(2\right) & X_{1}^{1}\left(2\right) & X_{2}^{1}\left(2\right) & X_{3}^{1}\left(2\right) & X_{4}^{1}\left(2\right)\\ Z_{1}\left(3\right) & X_{1}^{1}\left(3\right) & X_{2}^{1}\left(3\right) & X_{3}^{1}\left(3\right) & X_{4}^{1}\left(3\right)\\ \cdots & \cdots & \cdots & \cdots & \cdots \\ Z_{1}\left(k\right) & X_{1}^{1}\left(k\right) & X_{2}^{1}\left(k\right) & X_{3}^{1}\left(k\right) & X_{4}^{1}\left(k\right) \end{array}\right|$$
$$Y_{N}={\left|X_{0}^{0}\left(k\right)\right|}^{T}\cdots (k\geq 2)$$

The Matlab programming software (MathWorks. Inc., Natick, MA, USA) was utilized to calculate the Equation (6) to obtain a and b_i_, and then the influence of the content of each chemical component on the mechanical property could be judged through the grey differential equation of GM(1,4).
(6)$$\overset{ʌ}{a}={\left[\begin{array}{ccccc} a & b_{1} & b_{2} & b_{3} & b_{4} \end{array}\right]}^{\;T}={\left[B^{T}B\right]}^{\;-1}B_{T}Y_{N}$$

## 3. Results

### 3.1. Chemical Component Analysis

The contents of cellulose, hemicellulose, and lignin in 26 groups of samples were obtained by wet chemical analysis, three samples (No. 8, No. 23 and No. 26) were selected as an external validation set at random, and the remainder were used as a calibration set. The chemical components of 26 samples are shown in Table 2.

A larger coverage and dispersion of calibration set samples could make the model more capable. In the calibration set, the content of cellulose was from 38.02% to 76.97%; the hemicellulose was from 10.33% to 20.96%, while the lignin was from 7.93% to 28.24%, indicating that the content range of three components was extremely wide, which was conducive to the establishment of the GM. The standard deviations of cellulose, hemicellulose, and lignin in the calibration set were 8.34%, 2.62%, and 5.12%, respectively, which revealed that the component content of each sample was different. In other words, the samples were more dispersed. In addition, hemicellulose and lignin were removed during the degumming process, so the standard deviation was smaller than for cellulose. The cellulose content after degumming increased, the content coverage became wider, and the dispersion was larger. Therefore, the selected samples could be suitable for the establishment of the NIR prediction model.

### 3.2. Establishment of NIR Model

Spectra data was preprocessed, selecting the normalization, first-order derivative, and standard normal variate transformation (SNV). Therefore, the establishment of the NIR models was based on preprocessed spectral data, and optimized later.

The optimized parameters are listed in Table 3. The PLS factors (PCs) of cellulose, hemicellulose, and lignin prediction models were 5, 7, and 6, respectively, which indicated that the model was stable. In general, for all models, the values of RMSEC and RMSEP were less than 1, while the levels of Rc^2^ and Rcv^2^ were above 0.9, and RESEC values were less than those for RMSEP. All of them met the requirements of the modeling. The chemical component content obtained by chemical treatment in Table 2 was taken as the measured value of each sample, and the chemical component content predicted by the NIR model was taken as the predicted value. As shown in Figure 1, the measured values of the calibration set samples had a high correlation with the predicted values. Therefore, the three established NIR models showed high accuracy.

To further verify the accuracy and reliability of each NIR model, the validation set was applied for external validation, and SPSS software (IBM, New York, NY, USA) was used to perform a two-tailed *t*-test on the measured and predicted values of each sample in the validation set to evaluate the prediction results. As shown in Table 4, the absolute error of cellulose content was between 0.83% and 2.10%, the absolute error of hemicellulose content was between 0.34% and 0.4%, and the absolute error of lignin content was from 0.09% to 0.73%. Considering the wide content range and large dispersion of each component, these errors were relatively small, so the prediction accuracy of the NIR model was extremely high. When d_f_ = 2, α = 0.05, t_0_._05(2)_ = 4.303, the obtained results showed that three components were less than 4.303 (Table 5). This also shows that there were no significant differences between the measured values and the predicted values in the validation set. The results verified the reliability and accuracy of the NIR model, which could be initially applied to the determination of chemical components, laying a foundation for the rapid prediction of mechanical properties.

### 3.3. Young’s Modulus of WPF

WPF was tensile tested, as mentioned above. An analysis of the Young’s modulus of WPF was carried out using the system compliance method. The graphic representation of Equation (1) for sample No.1, tested at different gauge lengths, is shown in Figure 2. The specific parameters for all samples are displayed in Table 6. The value of Cs was small, in that the original data collected contains a higher linear region, which can reduce the error. The Young’s modulus was from 877.19 MPa to 2579.67 MPa in 15 samples, which indicated that it could be affected by the chemical degumming treatment. In general, the content of each chemical component of WPF affected the mechanical properties. Hemicellulose formed random, amorphous branched, or nonlinear structures, with little strength, and the molecular chain was shorter than that for cellulose [33,34]. Therefore, the decrease in hemicellulose was beneficial to the increase in the Young’s modulus. To further explore the influence of chemical components on the Young’s modulus, a GM was utilized to clarify the relationship between cellulose, hemicellulose, lignin, and the Young’s modulus.

### 3.4. Establishment of GM and Error Analysis

Using Equations (3)–(5), the one-time accumulated generating operation sequence and mean value generation sequence were acquired (Table 7). In addition, a and b_i_ of Equation (7) were obtained according to Equation (6) and substituted into Equation (2). Then the GM(1,4) model was established to explore the influence rule of chemical component contents of WPF on the Young’s modulus.
(7)$$\overset{ʌ}{a}=\left|\begin{array}{c} a\\ b_{1}\\ b_{2}\\ b_{3} \end{array}\right|=\left|\begin{array}{c} 0.3129\\ 0.5292\\ -0.9558\\ 1.1413 \end{array}\right|$$
(8)$$X_{0}^{\left(0\right)}\left(k\right)=0.5292X_{1}^{\left(1\right)}\left(k\right)-0.9558X_{2}^{\left(1\right)}\left(k\right)+1.1413X_{3}^{\left(1\right)}\left(k\right)-0.3129Z_{1}\left(k\right)$$

The GM(1,4) model of cellulose, hemicellulose, lignin content, and the Young’s modulus of WPF is shown in Equation (8). The positive values of grey action quantity *b*_1_ and *b*_3_ indicated that they had a positive effect on the increase in *X*_0_^(0)^(*k*); that is, the increase in cellulose and lignin content was beneficial to the increase in the Young’s modulus of WPF. On the contrary, the grey action quantity *b*_2_ was negative, which suggested that hemicellulose had an inhibitory effect on the Young’s modulus, and they were inversely proportional to each other in a certain range. The above analytical results were consistent with the theoretical study of each component. Cellulose is a strong, linear (crystalline) molecule, with no branching, and the interaction between molecular chains is very strong [35], which can form intermolecular hydrogen bonds, leading to the non-rotation of glycoside bonds, increasing their rigidity. Lignin is a complex and non-crystalline three-dimensional network of phenolic polymers that widely exists in higher plant cells, acting as a junction between cellulose and hemicellulose [36]. Therefore, lignin is able to bear external mechanical forces in its fiber. Consequently, the increase in cellulose and lignin content could improve the Young’s modulus of WPF, to a certain extent.

Error analysis was performed to verify the accuracy of the GM(1,4) model. Table 8 showed the error analysis between the original initialization processing data and initialization data, which was calculated by Equation (8).

The relative error of sample No.12 was 1.08, larger than for the other samples in Table 8. The reason was that the R^2^ of No.12 was 0.70 (Table 6), the correlation was less than for the other samples, which led to the larger error in the Young’s modulus. However, the average relative error was only 0.29, which indicated that GM(1,4) could be used to rapidly predict the WPF Young’s modulus through the chemical components. Therefore, in order to select suitable fibers for spinning, the NIR model could be used to quickly predict the content of each chemical component of WPF, and then the Young’s modulus could be predicted by the GM(1,4) model, which was beneficial to quickly and accurately select the fibers that met the spinning requirements.

## 4. Conclusions

In this work, the NIR model and the GM model were successfully established to achieve the rapid prediction of the mechanical properties of WPF. The acquired results show that, based on the quantitative analysis of cellulose, hemicellulose, and lignin of each sample using the wet chemical analysis method, the established NIR prediction model has high coefficients of determination (R_C_ and R_CV_) and low errors (RESEC < RMSEP). External validation exhibited excellent prediction accuracy, which indicated that the quantitative prediction of the model was possible. The effect of chemical component content on mechanical properties was investigated by using the grey system analysis theory, and then the GM(1,4) model for cellulose, hemicellulose, lignin, and the Young’s modulus was established. The obtained results showed that the content of cellulose and lignin could promote the increase in the Young’s modulus, while the increase in hemicellulose content inhibited it. This work provides a fast, accurate, and effective means for quantitative analysis of chemical components and the determination of physical properties, avoiding expensive, laborious, and time-consuming chemical analysis and physical testing. In addition, the two technologies of the NIR and GM models are in line with the trend of modernization and green development in the industry, which is conducive to the transformation, upgrading, and accelerated development of these enterprises. Therefore, this method also has great potential for the rapid and accurate testing of various other fibers.

## Figures and Tables

**Figure 1 materials-15-04989-f001:**
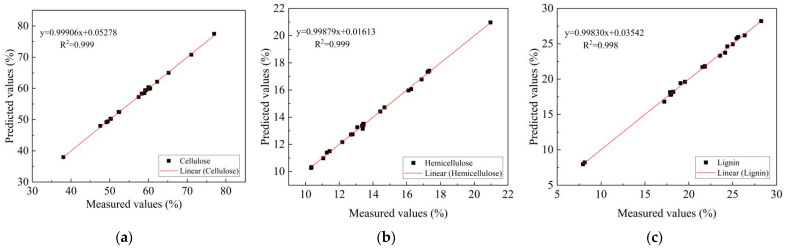
The measured values and predicted values of the calibration set samples: (**a**) cellulose, (**b**) hemicellulose, (**c**) lignin.

**Figure 2 materials-15-04989-f002:**
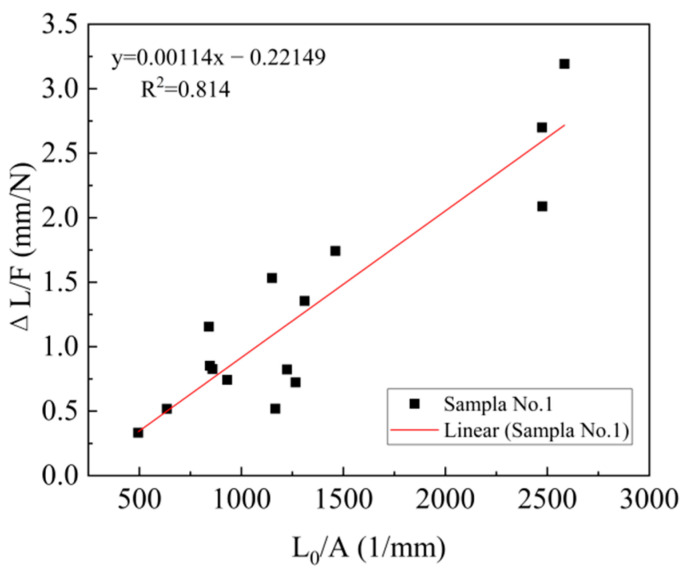
System compliance method for sample No.1.

**Table 1 materials-15-04989-t001:** The chemical components of several common WPF.

Chemical Component (%)	Component Content			
	Leaf Sheaths	Trunks	Fruit Bunch	Mesocarp
cellulose	22.35 ± 0.31	44.25 ± 2.96	40.50 ± 0.37	32.00~60.00
hemicellulose	41.30 ± 2.80	33.94 ± 1.25	24.30 ± 0.14	9.80~31.00
lignin	36.35 ± 2.89	33.12 ± 2.16	35.20 ± 0.11	11.00~32.80

**Table 2 materials-15-04989-t002:** Chemical components of the windmill palm.

Sample	Cellulose (%)	Hemicellulose (%)	Lignin (%)
1	25.63	12.79	49.12
2	28.24	10.33	50.27
3	25.44	10.35	50.31
4	25.01	11.27	52.46
5	23.56	12.18	46.33
6	21.81	11.43	60.44
7	19.06	13.39	60.01
8	57.67	13.19	20.66
9	24.14	13.42	52.28
10	17.89	13.40	65.20
11	26.36	13.35	50.16
12	17.86	12.69	59.31
13	24.39	13.07	47.55
14	21.55	14.68	57.45
15	19.56	16.26	59.97
16	18.24	17.26	58.25
17	17.20	16.88	62.24
18	17.89	17.24	59.14
19	18.19	16.10	60.30
20	17.98	17.33	58.97
21	21.83	20.96	49.49
22	26.37	11.05	38.02
23	37.93	12.23	30.82
24	8.13	13.45	76.97
25	7.93	14.43	71.06
26	80.58	14.12	3.94

**Table 3 materials-15-04989-t003:** The correction model results.

Chemical Component	PCs	RMSEC	Rc^2^	RMSEP	Rcv^2^
cellulose	5	0.25	0.99	0.9	0.98
hemicellulose	7	0.09	0.99	0.95	0.93
lignin	6	0.21	0.99	0.89	0.97

**Table 4 materials-15-04989-t004:** Comparison between measured values and predicted values of validation set samples.

Sample	Cellulose			Hemicellulose			Lignin		
	Measured Values (%)	Predicted Values (%)	Absolute Error (%)	Measured Values (%)	Predicted Values (%)	Absolute Error (%)	Measured Values (%)	Predicted Values (%)	Absolute Error (%)
8	57.67	58.88	1.21	13.19	13.53	0.34	20.66	21.10	0.44
23	37.93	38.76	0.83	12.23	11.83	0.4	30.82	31.55	0.73
26	80.58	78.48	2.10	14.12	14.46	0.34	3.94	3.85	0.73

**Table 5 materials-15-04989-t005:** Results of two-tailed *t*-test.

	Cellulose	Hemicellulose	Lignin
|t|	0.019	0.378	1.500

**Table 6 materials-15-04989-t006:** Young’s Modulus of windmill palm fiber using the system compliance method.

Sample	Line Equation	*Cs* (mm/N)	*E* (MPa)	R^2^
1	Δ*L*/*F* = 0.00114 *L*_0_/*A* − 0.22149	−0.22149	877.19	0.81
2	Δ*L*/*F* = 7.2952 × 10^−4^ *L*_0_/*A +* 0.14939	0.14939	1370.75	0.77
3	Δ*L*/*F* = 8.29946 × 10^−4^ *L*_0_/*A* + 0.04912	0.04912	1204.90	0.90
4	Δ*L*/*F* = 3.44822 × 10^−4^ *L*_0_/*A* + 0.36164	0.36164	1900.05	0.72
5	Δ*L*/*F* = 5.32816 × 10^−4^ *L*_0_/*A* + 0.15288	0.15288	1876.82	0.95
6	Δ*L*/*F* = 6.7353 × 10^−4^ *L*_0_/*A* + 0.26701	0.26701	1484.71	0.76
7	Δ*L*/*F* = 3.00383 × 10^−4^ *L*_0_/*A* + 0.46724	0.46724	2529.08	0.92
8	Δ*L*/*F* = 7.32367 × 10^−4^ *L*_0_/*A* + 1.17335	1.17335	1365.44	0.74
9	Δ*L*/*F* = 4.30302 × 10^−4^ *L*_0_/*A* + 0.43854	0.43854	1323.95	0.78
10	Δ*L*/*F* = 3.87646 × 10^−4^ *L*_0_/*A* + 0.69731	0.69731	2579.67	0.74
11	Δ*L*/*F* = 8.69669 × 10^−4^ *L*_0_/*A* + 0.15684	0.15684	1149.86	0.74
12	Δ*L*/*F* = 7.73499 × 10^−4^ *L*_0_/*A* − 0.04913	−0.04913	1292.83	0.70
13	Δ*L*/*F* = 8.98708 × 10^−4^ *L*_0_/*A* − 0.07606	−0.07606	912.71	0.79
14	Δ*L*/*F* = 3.5204 × 10^−4^ *L*_0_/*A* + 0.13491	0.13491	1840.59	0.86
15	Δ*L*/*F* = 5.48148 × 10^−4^ *L*_0_/*A* + 0.21965	0.21965	1824.32	0.75

**Table 7 materials-15-04989-t007:** The values of cumulative and mean generating sequences of samples 1–15.

Sample	*X*_1_^(1)^(*k*)	*X*_2_^(1)^(*k*)	*X*_3_^(1)^(*k*)	*X*_0_^(1)^(*k*)	*Z*_1_(*k*)
1	1.00	1.00	1.00	1.00	——
2	2.02	1.81	2.10	2.56	1.78
3	3.05	2.62	3.09	3.94	3.25
4	4.12	3.50	4.07	6.10	5.02
5	5.24	4.45	4.99	8.24	7.17
6	6.47	5.34	5.84	9.93	9.09
7	7.69	6.39	6.58	12.82	11.38
8	8.90	7.50	7.39	14.37	13.60
9	9.96	8.55	8.33	15.88	15.13
10	11.29	9.60	9.03	18.82	17.35
11	12.31	10.64	10.05	20.14	19.48
12	13.52	11.63	10.75	20.61	20.37
13	14.49	12.65	11.70	22.65	21.63
14	15.66	13.80	12.54	24.75	23.70
15	16.82	15.15	13.25	26.83	25.79

**Table 8 materials-15-04989-t008:** Error analysis of GM(1,4) model.

Sample	*X*_0_^0^(*k*)′	*X*_0_^0^(*k*)	Relative Error
1	1.18	1.56	0.24
2	1.63	1.37	0.18
3	1.91	2.17	0.12
4	1.97	2.14	0.08
5	2.14	1.69	0.26
6	1.92	2.88	0.33
7	1.72	1.56	0.10
8	1.87	1.51	0.24
9	1.67	2.94	0.43
10	1.73	1.31	0.32
11	1.93	1.47	0.31
12	2.16	1.04	1.08
13	1.99	2.10	0.05
14	1.48	2.08	0.29
15	1.18	1.56	0.24
average relative error: 0.29			

## Data Availability

The data presented in this study are available on request from the corresponding author.

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
