# Peer review of "Rapid Prediction of Mechanical Properties Based on the Chemical Components of Windmill Palm Fiber"

_materials, 2022, doi:10.3390/ma15144989_

Round 1

Reviewer 1 Report

1. In the introduction section the GA for establishing the features of other natural fibers be introduced.

2. The ingredient of WPF like cellulose, hemicellulose and lignin, which were reported by various researches, be mentioned in form of table into the introduction section.

3. It is recommended that the GA with other methods, which were used for establishing the mechanical properties of WPF, be compared in form of Table.

4. The results and discussion section can be reinforced with the proper references.   

Reviewer 2 Report

The authors developed a grey model (GM) for rapid and accurate prediction of the mechanical property of windmill palm fiber (WPF) to explore the effect of chemical component content on the Young's modulus. The chemical component content of cellulose, hemicellulose and lignin in WPF was studied by near-infrared (NIR) spectroscopy, and NIR prediction model was established with the chemical measured values as the control. The value of RC and RCV was more than 0.9, while RMSEC and RMSEP were less than 1, which reflected the excellent accuracy of NIR model. External validation and two-tailed t-test were used to evaluate the accuracy of NIR model prediction results. The GM (1,4) model of WPF chemical component and Young's modulus was established. GM (1,4) indicated that cellulose and lignin content promoted Young's modulus, and hemicellulose content inhibited breaking strength. The establishment of the two models provides a theoretical basis for evaluating whether WPF can be used in spinning, which is convenient for the selection of spinning fibers in practical application.

The paper is well written and could be published after major revision.

1. What is the basis for the experimental design? What is the advantage of NIR spectra?

2. The abstract should describe some experiment results.

3. The absolute error in Table 3 should be added.

4. Why the chemical component of windmill palm was so different in table 1, as NO.8, 24, 25 and 26.

5. How to select the cross-section of the fiber in the Young's modulus measurement?

6. The English style and writing need to be improved.

7. “The effect of chemical component content on properties was investigated by using the grey system analysis theory, and GM (1,4) model of cellulose, hemicellulose, lignin as well as Young's modulus was established.” Clarify?

8. “ The grey action quantity of cellulose and lignin were positive, which indicated that the two components could promote the increase of Young's modulus of WPF.” Why?

Reviewer 3 Report

The manuscript presented by Guan et al. presents a method to study the effect of the composition of windmill palm fiber on the mechanical properties of these bio-materials. The study shows interesting strategies to characterize these fibers and could be of potential interest for the scientific community using these as matrix/reinforcement of bio-based materials. However, there is room for improvement of the study and some motivation and validations are needed to lift the impact of the manuscript. Here I detailed some suggestions:

1) Line 26: what is meant by promoted Young modulus? Line 27: what is meant by inhibited? 

2) Line 34-35: please, include some significant values on the windmill palm production. For example, production size, advantages over other crops, etc.

3) Line 42: the chemical component content of WPF has been studied before (https://doi.org/10.3390/f10121130, doi.org/10.1177/1558925019883451). The authors should lift the novelty of the work to increase the impact. 

4) Lines 44-45: Please, provide examples of which are the "traditional methods" for determining the composition of these crops. 

5) Line 51: Please, introduce the background of gray models (GM) when first discussed in the introduction. 

6) What is the advantage of this proposed method over other prediction models (e.g., neuronal networks) and/or simulation pathways? See: https://doi.org/10.1016/j.matdes.2007.02.009

7) What are the future prospects of this methodology and its applicability? Please, clarify.

8) Line 101: How were the samples dried? How many replicates were tested in the NIR experiments? 

9) Section 2.5: how were the samples conditioned? How were the samples tested? as individual fibers? Please, clarify.

10) Line 139: what are these 26 groups of samples? 

11) Line 194: which chemical treatment are the authors referring to? 

12) How is the data presented correlated to other "destructive" methods? There is no comparison of how this methodology compared to other traditional methods, which is the key component of this study, i.e., proposing a new "non-destructive" pathway to characterize these materials. 

Reviewer 4 Report

Dear Authors,

Generally, this research work has a fairly narrow scope of research. This is not an advantage of this research paper. In my opinion, the introduction could also be written a bit more. I have a similar impression when describing and discussing the research results. After all, I must admit that I really like the work.

To improve the quality of this work a little, here are some detailed comments:

Line 50: Currently, more and more work is related to research on various biocomposites made from plant materials. In this case, the quick determination of the chemical composition of the raw material such as lignin, cellulose and hemicellulose may be of key importance for the repeatability of production. In addition, as you have noticed, it has a large impact on the strength of products made of plant materials. After all, lignocellulose is the main component of many by-products of agri-food processing. Therefore, it is worth extending this description and providing other examples (possibilities) of your research applications. In this or that place. Browse through sample articles of such studies: "Properties of Biocomposites Produced with Thermoplastic Starch and Digestate: Physicochemical and Mechanical Characteristics", "Properties of Biocomposites from Rapeseed Meal, Fruit Pomace and Microcrystalline Cellulose Made by Press Pressing: Mechanical and Physicochemical Characteristics", "Comparison of some biocomposite board properties fabricated from lignocellulosic biomass before and after Ionic liquid pretreatment ”.

Line 71: If possible, add information on how much time you can save using the NIR method (eg how long it takes to test one sample). Also, write down what values ​​of the time reduction industry can expect. Then present this information against the background of the standard methods. This will highlight the purpose of your research. Additionally, describe the other benefits of using this method.

Line 74 - 78: This piece of work looks like information from an abstract or research methodology. This excerpt can be placed elsewhere in the paper (possibly clearly indicate that this is the scope of the research). In my opinion, however, a better solution would be a short description of the utilitarian goal.

Line 139: The software must also be precisely described (name: manufacturer, city, country).

Line 141: Justify why you selected exactly 26 samples. On what basis did you make such a choice.

Line 161: "relatively stable" is not a good term for a model. This should be clearly stated.

Line 174: The methodology lacks information on the use of two-sided t-tests. Moreover, what software was used for this purpose?

Line 240. The conclusions are correctly written. However, one "perspective" conclusion can be added, taking into account other possible applications of your model.

Round 2

Reviewer 2 Report

The paper is excellent and ready for publication after major revision.

But equations 1-5 need reference.

The introduction section should be improved and recent papers should be cited especially from mdpi.

How did you obtain regression equation in QQ plots?

A schematic of the model should be included.

Reviewer 3 Report

The revised manuscript has answered the majority of the comments raised by the reviewers. Hence, it can be accepted in the present form.

Author Response

Thank you for the decision letter and for everything you have done to improve our manuscript. Moreover, we highly appreciate the evaluation and suggestions in our manuscript.

Reviewer 4 Report

Dear Authors,

Thank you for adapting to my comments. In general, the introduction can be refined a bit, but it's still good. I accept the corrections made.

Author Response

(The authors gave the same response as above.)
